# Coherent X-ray measurement of step-flow propagation during growth on polycrystalline thin film surfaces

Randall L. Headrick [1], Jeffrey G. Ulbrandt[1], Peco Myint [2], Jing Wan[1], Yang Li[1], Andrei Fluerasu[3], Yugang Zhang[3], Lutz Wiegart[3] & Karl F. Ludwig Jr.[2,4]

The properties of artificially grown thin films are strongly affected by surface processes during growth. Coherent X-rays provide an approach to better understand such processes and fluctuations far from equilibrium. Here we report results for vacuum deposition of $C_{60}$ on a graphene-coated surface investigated with X-ray Photon Correlation Spectroscopy in surface-sensitive conditions. Step-flow is observed through measurement of the step-edge velocity in the late stages of growth after crystalline mounds have formed. We show that the step-edge velocity is coupled to the terrace length, and that there is a variation in the velocity from larger step spacing at the center of crystalline mounds to closely-spaced, more slowly propagating steps at their edges. The results extend theories of surface growth, since the behavior is consistent with surface evolution driven by processes that include surface diffusion, the motion of step-edges, and attachment at step edges with significant step-edge barriers.

[1] Department of Physics and Materials Science Program, University of Vermont, Burlington, VT 05405, USA. [2] Division of Materials Science and Engineering, Boston University, Boston, MA 02215, USA. [3] National Synchrotron Light Source II, Upton, NY 11967, USA. [4] Department of Physics, Boston University, Boston, MA 02215, USA. Correspondence and requests for materials should be addressed to R.L.H. (email: rheadrick@uvm.edu)

Studies of thin film growth seek to understand the dynamics of surface nanostructures that are simultaneously undergoing both stochastic particle deposition and random relaxation processes. These processes, which affect surface structure, morphology and composition[1] as well as defect propagation[2] play a central role in determining the properties of artificially grown thin films[3–5]. However, traditional methods used to study growth in real-time such as electron diffraction[6,7] and surface X-ray scattering[8,9] are unable to provide a complete understanding of surface dynamics since they suffer from the limitation that the surface must be an almost perfectly flat single crystal. For example, observations of layer-by-layer oscillations yield kinetic information from which energy barriers to surface diffusion and interlayer transport can be deduced, but only during the early stages of growth before significant roughness develops. Although nucleation and coalescence continue locally, layer-by-layer growth oscillations become unobservable in the presence of surface roughness since they are damped out by phase differences in the scattering from different regions of the surface.

Recent advances in coherent X-ray methods that utilize X-ray Photon Correlation Spectroscopy (XPCS)[10] can yield crucial information on the dynamics where the structural fluctuations about an average configuration occur. This is possible since the scattering of coherent X-rays produces a speckle pattern, which depends sensitively on the detailed configuration within each coherence volume. The XPCS analysis characterizes the time correlations of fluctuations in equilibrium or out-of-equilibrium systems. Examples include step-edge fluctuations during annealing and evaporation[11,12], surface roughness fluctuations during deposition[13], fluctuations at polymer surfaces[14], and the dynamics of bulk phase transformations[15,16]. These fluctuations are invisible to analysis with low-coherence X-rays, since those methods average over local structures within the illuminated volume. Coherent X-rays can also be used to measure surface velocities where heterodyne mixing between scattering from different regions mix to produce oscillatory correlations. For example, defect velocity[17], has been studied during growth of amorphous thin films by sputter deposition.

Here, we show that it is possible to utilize coherent X-rays to study thin film growth without loss of information due to spatial averaging, even in the later stages where the growth surface is very rough, and the film is composed of many separate crystalline grains. Step flow processes are observable by correlation spectroscopy when there is no change of X-ray intensity as the steps advance, and hence layer-by-layer oscillations are not present. The effects are analogous to oscillations in homodyne correlations that have been observed under flow conditions or during elastic relaxation, which can be observed if there is a velocity gradient[18–21]. The measurements show promise for making detailed comparisons with theories of crystal growth.

Figure 1a shows an overall schematic of the thin film deposition by thermal evaporation of solid $C_{60}$ onto an amorphous substrate in a vacuum environment to form a polycrystalline thin film. We follow the surface evolution from the initial nucleation stage through to steady-state growth, focusing on the properties and dynamics of the steady-state regime. A coherent X-ray beam is incident on the substrate surface during the growth, and scattered X-rays are detected by a fast area detector (see Methods for further details). Figure 1a also shows an example of surface topography obtained by post-deposition Atomic Force Microscopy, as well as a speckle pattern from the Grazing Incidence Small Angle X-ray Scattering pattern acquired during the growth. The close-up inset in the speckle pattern corresponds to a small region of the scattering pattern near the Yoneda wing, which is visible as a horizontal streak in the main image. It is due to an enhancement of the surface

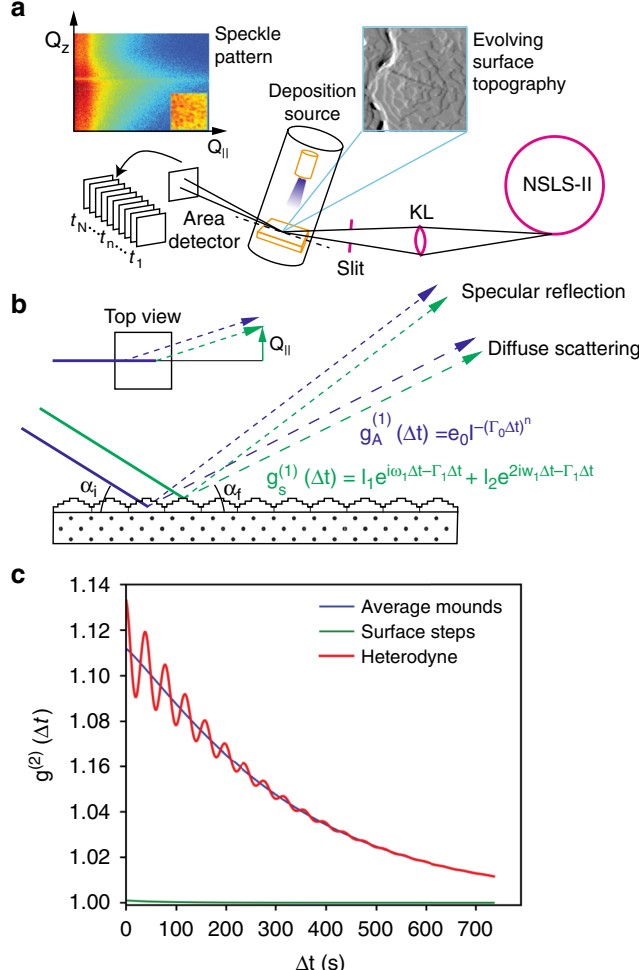

**Fig. 1** Schematic of the experiment and coherent mixing effects during $C_{60}$ thin film deposition. **a** X-rays from the synchrotron source are focused by a kinoform lens (KL) and a collimating slit system into an ultra-high vacuum sample enclosure. A polycrystalline thin film is deposited, which causes (111)-oriented crystalline mounds to form via nucleation at the top of each mound and local step-flow towards the mound edges. Scattered coherent X-rays form speckle patterns that correspond to the detailed configuration of the surface and are recorded versus time by a high-resolution photon sensitive X-ray area detector. **b** In addition to scattering from the surface (green lines and equation), the X-rays scatter from the average mounds structure (blue). The functions $g_A^{(1)}(\Delta t)$ and $g_s^{(1)}(\Delta t)$ correspond to the intermediate scattering functions for the average and surface mound contributions respectively. **c** The two signals interfere coherently, creating temporal correlations in the speckle pattern that can oscillate with the frequency $\omega_1$, which is directly related to the step-edge velocity. This effect occurs even when the averaged intensity is nearly static. The second-order correlation function $g^{(2)}(\Delta t)$ is extracted from intensity data, as described in the main text

diffuse scattering at exit angles $\alpha_f$ near the critical angle for total external reflection ($\alpha_c \approx 0.16°$ for 9.65 keV X-rays incident on solid $C_{60}$). Figure 1b, c illustrates the principle of heterodyne mixing, where in this case strong scattering from the average configuration of surface mounds mixes coherently with the much weaker scattering from surface steps. As a result, oscillations are observed due to the motion of the steps across the surface, and the amplitude of the oscillation is much larger than the scattering from the steps separately. As we discuss in more detail below, valuable insight is obtained about the step motion

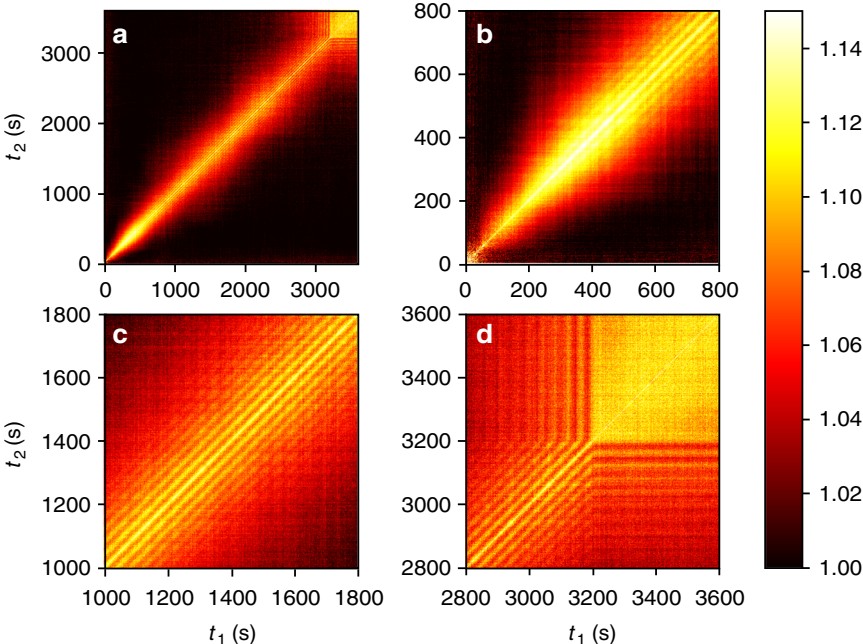

**Fig. 2** Two-time correlations for $C_{60}$ deposition on graphene/$SiO_2$. **a** The complete 1-h data collection during deposition with a substrate temperature of 144 °C. The deposition shutter was opened at 40 s. after the start of the scan and closed again at 3200 s. **b** detailed view of the early time island nucleation and transition to local step-flow growth. **c** close-up during the middle of the scan with stationary dynamics. **d** view of the end of the deposition, showing that the dynamics stop abruptly when the growth shutter is closed. The data was collected at $Q_{||} = 0.0115$ Å$^{-1}$ and $Q_z = 0.045$ Å$^{-1}$

in this experiment. Molecular steps do not move all in the same direction or at the same rate; instead, they slow down as they approach the edge of mounds where the terrace length becomes smaller.

### Results

**Correlations in high temperature growth.** The non-stationary dynamics during surface growth is measured via two-time intensity autocorrelation functions, which are derived from experimental X-ray intensities $I(\mathbf{Q}, t)$[15,22]:

$$G(\mathbf{Q}, t_1, t_2) = \frac{\langle I'(\mathbf{Q}, t_1) I'(\mathbf{Q}, t_2) \rangle_Q}{\langle I'(\mathbf{Q}, t_1) \rangle_Q \langle I'(\mathbf{Q}, t_2) \rangle_Q}. \quad (1)$$

The normalized intensity is obtained from $I'(\mathbf{Q}, t) = I(\mathbf{Q}, t)/\widetilde{I(\mathbf{Q})}$, where $\widetilde{I(\mathbf{Q})}$ is averaged over time and over a few detector pixels to smoothen out the speckles. The average $\langle \rangle$ is obtained by averaging over a range of $\mathbf{Q}$ having similar time correlations (Supplementary Note 1).

Figure 2 shows the two-time correlations $G(\mathbf{Q}, t_1, t_2)$ for deposition at $T_{sub} = 144$ °C. Figure 2a shows the complete deposition. Figure 2b shows two-time correlations from the start of deposition at $t = 40$ s and during the period of roughening where mounds initially form and stabilize. The most striking feature of the data is the transition to a pattern of parallel streaks that appear between 400 and 600 s (~9–13 monolayers). Layer-by-layer oscillations have previously been observed for deposition of $C_{60}$ on mica at 60 °C in the early stages of deposition[23]. However, we note that the oscillations we observe have a different characteristic than the layer-by-layer oscillations, which occur at the beginning of the growth process before the surface becomes rough. In contrast, the oscillations in the correlations observed in Fig. 2 emerge during 3D growth and persist for the remainder of the deposition without any discernable decay (Fig. 2c), indicating that they are part of the steady-state dynamics. These streaks cease when the growth shutter is closed

at 3200 s (Fig. 2d). Correlations peak when $t_1 = t_2$, and decay as a function of $\Delta t = t_1 - t_2$ over the entire time interval.

For steady-state dynamics, the one-time correlation function can be employed:

$$g^{(2)}(\mathbf{Q}, \Delta t) = \frac{\langle I(\mathbf{Q}, t) I(\mathbf{Q}, t + \Delta t) \rangle_t}{(\langle I(\mathbf{Q}, t) \rangle_t)^2}. \quad (2)$$

Figure 3a shows correlations for the same deposition shown in Fig. 2, averaged over the time interval from 650 s to the end of the deposition. The data exhibits pronounced oscillations as a function of $\Delta t$ with a peak in amplitude at an in-plane wave-vector of $Q_{max} = 0.0145$ Å$^{-1}$, corresponding to a length scale of ≈43 nm. The period of the oscillations is $T = 40$ s, and it does not shift with $Q_{||}$. The value calculated from the deposition rate (1.11 nm/min) and (111) layer spacing is $T_{dep} = 43$ s. Therefore, the period of oscillations is very close to the monolayer deposition time for $C_{60}$ growth. Several additional features of the data can be seen more clearly in Fig. 3b, which is for a thin film deposited with a substrate temperature of $T_{sub} = 216$ °C. The maximum oscillation amplitude is at $Q_{max} = 0.0050$ Å$^{-1}$, which corresponds to a length scale of 126 nm. We observe a general trend that the maximum oscillations occur at a value of $Q_{||}$ that shifts lower, corresponding to a larger length scale for different samples, as the temperature is increased (Supplementary Note 2). The curve at $Q_{||} = 0.0085$ Å$^{-1}$ exhibits sharper maxima and broader minima, with weak maxima corresponding to correlations with a half-monolayer period.

Figure 4 shows post-growth characterization of one of the samples. Figure 4a shows an Atomic Force Microscope (AFM) scan of the film surface where the grain structure is visible. Molecular steps are also observed in the image. They are roughly circular, surrounding a high point near the center of each mound, and the mean step spacing on the tops of the mounds for growth at 216 °C is estimated to be ≈150 nm. This value can be compared to the length scale ≈126 nm derived from $Q_{max}$ for the data in Fig. 3b and Supplementary Fig. 2e. The full amplitude mode and

height mode images corresponding to the inset in Fig. 4a are shown in Supplementary Note 3. Figure 4b shows an X-ray specular scan. Only (111) and (222) reflections are observed, indicating that the films are highly oriented.

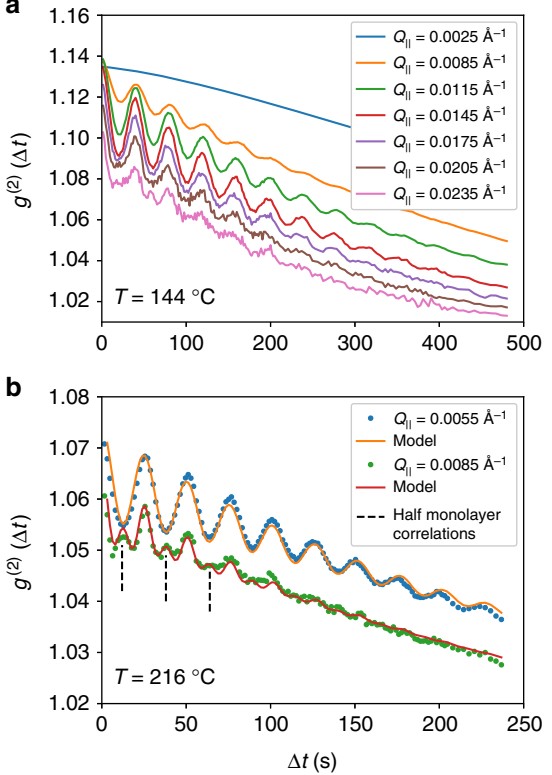

**Fig. 3** Autocorrelations during steady-state deposition. **a** Growth at 144 °C as a function of $Q_\parallel$, with fixed $Q_z = 0.045$ Å$^{-1}$. Note that the oscillation frequency is independent of $Q_\parallel$. The step spacing can be estimated from $2\pi/Q_{max}$, where $Q_{max}$ is the in-plane component of the wave-vector transfer at which $g^{(2)}$ has the maximum oscillation amplitude. **b** Growth at 216 °C for two different $Q_\parallel$. The shape of the autocorrelation curve changes at large $Q_\parallel$, where the maxima become sharp and the minima become broadened. A half-monolayer correlation is observed, which is indicated by dashed lines. An empirical model to fit the data is described in the main text. Note that the upper curve in **b** is offset by 0.01 for the sake of clarity

The one-time correlation function (Eq. 2) can be decomposed into a simpler product of correlation functions of electric fields rather than intensities:

$$g^{(2)}(\mathbf{Q}, \Delta t) = 1 + \beta(\mathbf{Q})|F(\mathbf{Q}, \Delta t)|^2 \qquad (3)$$

where $F(\mathbf{Q}, \Delta t) = g^{(1)}(\mathbf{Q}, \Delta t)/g^{(1)}(\mathbf{Q}, 0)$ is the normalized intermediate scattering function with $g^{(1)}(\mathbf{Q}, \Delta t) \sim \langle E(\mathbf{Q}, t')E^*(\mathbf{Q}, t' + \Delta t)\rangle_{t'}$, and $\beta(\mathbf{Q})$ is the optical contrast factor[24,25]. The intermediate scattering function is related to density-density variations in the sample, and in the case of Grazing Incidence Small Angle X-ray Scattering (GISAXS) the surface scattering is related to variations in the height of the surface through $g_s^{(1)}(\mathbf{Q}, \Delta t) \sim \langle h(Q_r, t')h^*(Q_r, t' + \Delta t)\rangle$. The experimental results suggest that the statistical properties of the growing surface can be described by an empirically-derived intermediate scattering function of the form:

$$g^{(1)}(\mathbf{Q}, \Delta t) = I_0 \exp\{-(\Gamma_0 \Delta t)^n\} + I_1 \exp\{i\omega_1 \Delta t - \Gamma_1 \Delta t\} + I_2 \exp\{2i\omega_1 \Delta t - \Gamma_1 \Delta t\} \qquad (4)$$

where $\omega_1$ is the oscillation frequency, and $\Gamma_j(Q_\parallel) = 1/\tau_j(Q_\parallel)$ is the relaxation rate, or inverse of the relaxation time. This form matches closely to the heterodyne form describing capillary waves on liquid surfaces[26], with one notable difference: for capillary waves, the frequency is proportional to the in-plane component of the wave vector transfer through the relation $\omega = Q_\parallel v$ where $v$ is the wave velocity, while in the present case the frequency is entirely independent of the wave-vector transfer. Instead, it is related to the monolayer deposition time by $\omega_1 = 2\pi/T_{dep}$ for all $Q_\parallel$. We also introduce a harmonic frequency $2\omega_1$ in the last term in Eq. 4 to take into account the presence of the observed half-monolayer correlations in the data. The stretching exponent $n$ in the first term takes into account the deviation of the overall decay from a simple exponential shape. Figure 3b includes curves fitted to the correlation results using Eqs. 3 and 4, and good agreement is obtained. Additional fitting results for the correlations shown in Fig. 3a are shown in Supplementary Note 4.

These results present a puzzle for the interpretation of the observed oscillations, since they suggest that we are not simply measuring the mean velocity of a uniformly propagating array of steps. A propagating array of steps does not by itself produce oscillatory correlations, due to translational symmetry. Only the phase advances, which is not directly observable. A quasi-static reference signal can mix with the scattering from the moving

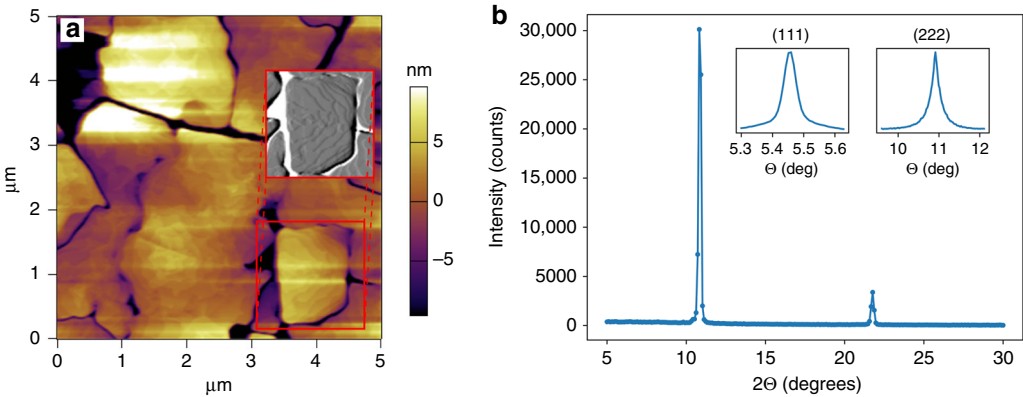

**Fig. 4** Post-deposition characterization of a C$_{60}$ thin film. **a** AFM image of the surface of a thin film growth in two layers with a sample temperature of 196 and 216 °C for the first and second layers respectively. The total film thickness is 198 nm. The inset shows a portion of the image plotted in amplitude mode, which makes the molecular terraces more visible. **b** X-ray diffraction characterization of the same sample. The main plot shows a $\theta - 2\theta$ scan through the (111) and (222) reflections. The insets show $\theta$-rocking scans through each reflection

steps, which would produce oscillations. In this case, a possible origin of such a quasi-static reference is the large-scale mound features present on the surface. Alternately, a layer-by-layer growth mode will produce oscillations with a frequency that is independent of $Q$. However, it is the surface roughness that oscillates in that case, so there will be a corresponding oscillation in the diffuse scattering intensity, which we do not observe (Supplementary Note 5). Moreover, layer-by-layer growth with a significant Ehrlich-Schwoebel step-edge barrier leads to an unstable surface where the surface quickly becomes too rough for the topmost layer to reach completion before nucleation of the next layer[27]. Layer-by-layer oscillations are therefore incompatible with our observation of correlation oscillations that continue in the late stages of growth without significantly decreasing in amplitude.

We emphasize that it is possible to observe oscillatory correlations in a coherent scattering measurement without corresponding oscillations in the intensity. This occurs when individual speckles oscillate in time, but since there is no simple relationship in the temporal phases from one speckle to the next for scattering from a complex surface structure, averaging over a region of interest that contains hundreds of speckles leads to the intensity oscillations being averaged out. On the other hand, the correlation functions in Eqs. 1 and 2 correlate different times before the Q-averaging is performed. This preserves the temporal information so that peaks in the correlations are observed for time differences $\Delta t$ where the surface reaches a self-similar state.

**Step flow model.** In order to improve on the models described above, we introduce the assumption that the steps are not uniformly spaced. Instead, the $n$th terrace has a variable length $L_n$, and the steps move at velocity $v_n$ rather than at a single overall velocity (Fig. 5). The oscillation period is interpreted as the time for surface steps to advance by one terrace length. Within this model, the presence of oscillatory correlations is intuitive in the sense that the surface returns to a self-similar state each time the terraces advance by one terrace length, for time intervals that are integer multiples of the monolayer deposition time $T_{dep}$.

This model readily incorporates mounds, and we refer to it as the Local Step Flow (LSF) model since steps are confined to each crystalline mound. The terrace length $L_n$ and step velocity $v_n$ are proportional to each other as the mound configuration approaches a steady state[28–30]. Hence, the period $T_{LSF} = L_n/v_n$ corresponds to the time for steps to advance by one terrace length as we require, but $T_{LSF}$ is itself independent of $n$. As a result, there is only a single oscillation frequency $\omega_{LSF}$.

The LSF model fully embodies the properties needed to describe the results of Fig. 3. The first term in Eq. 4 is interpreted as arising from the time-averaged mound structure, which reaches a nearly steady-state, or quasi-static configuration, as the film deposition proceeds. The second term is from a part of the nonuniform array of steps with an average spacing that matches $2\pi/Q_\parallel$ for a certain $Q_\parallel$. The frequency is set by the monolayer completion time $T_{LSF}$, as discussed above. The same segment of steps can also generate a (weak) doubled frequency at higher $Q_\parallel$, corresponding to a 2nd order reflection from the step array, which accounts for the third term on the rhs of Eq. 4. The combined signal is considered to be in a heterodyne mode since it consists of the very strong quasi-static term (average mounds) that mixes with a weaker scattering from the steps.

In order to study the LSF model, we employ a model for molecular beam epitaxy in $1 + 1$ dimensions previously introduced by Politi and Villain, called the Zeno model[31]. In the presence of an energy barrier that hinders interlayer diffusion, molecules landing on terraces cannot easily step down

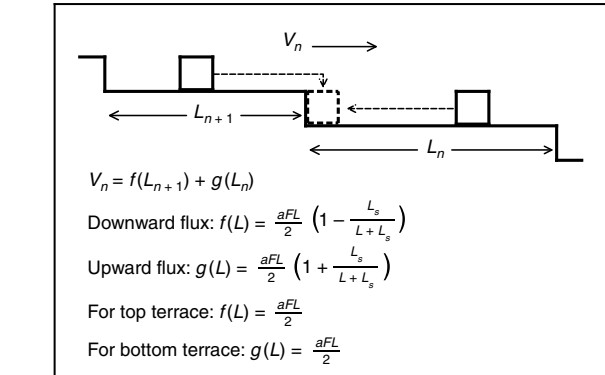

**a**

$$V_n = f(L_{n+1}) + g(L_n)$$

Downward flux: $f(L) = \dfrac{aFL}{2}\left(1 - \dfrac{L_s}{L + L_s}\right)$

Upward flux: $g(L) = \dfrac{aFL}{2}\left(1 + \dfrac{L_s}{L + L_s}\right)$

For top terrace: $f(L) = \dfrac{aFL}{2}$

For bottom terrace: $g(L) = \dfrac{aFL}{2}$

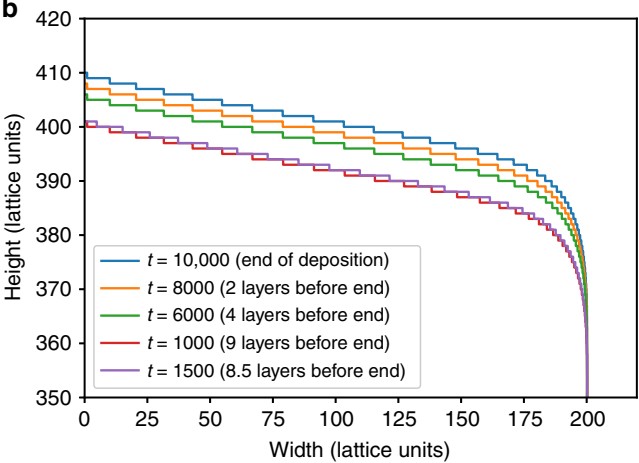

**b**

**Fig. 5** Schematic diagram of the step-flow growth model. **a** In the Zeno model, the velocity of step $n$ is determined by the widths of the upper ($L_{n+1}$) and lower ($L_n$) terraces. Ad-molecules diffusing on the upper terrace attach to the step-edge by descending from above, while those below attach directly. **b** The propagation of steps on the surface is shown at various times during the growth, starting with the top terrace at a height of 400 lattice units, and ending when the top terrace nucleates at a height of 410 lattice units. $L_s$ is equal to 4. The Zeno model is explained in the main text

(the Ehrlich-Schwoebel effect)[32,33]. Since grooves at the boundary between step arrays are not easily filled up, three-dimensional islands or mounds are formed that have a wedding cake type structure. In the present case, mounds are practically guaranteed to form due to the polycrystalline nature of the $C_{60}$ films. This is due to the fact that the islands forming the base layer of mounds nucleate without any preferred azimuthal orientation, so neighboring mounds are inhibited from merging as they impinge, ensuring that deep grooves develop at their boundaries. We note that a similar mechanism has previously been suggested to explain rapid roughening in diindenoperylene thin films with tilt domain boundaries[34]. Given these considerations, a model that incorporates groove formation seems particularly apt. The model introduced by Politi and Villain is known as the Zeno model since the steps slow down as they approach the edges of the mound. Nucleation of new terraces occurs only at the top of the mounds[35], while steps propagate towards the mound edges. The model is entirely deterministic since nucleation always occurs at the exact center of the layer beneath it, at the moment when that step reaches the critical length for nucleation.

Figure 5 shows the terrace structure of a mound generated with the Zeno model. The model incorporates the Ehrlich-Schwoebel effect through a single parameter, $L_s$, the Schwoebel length, which

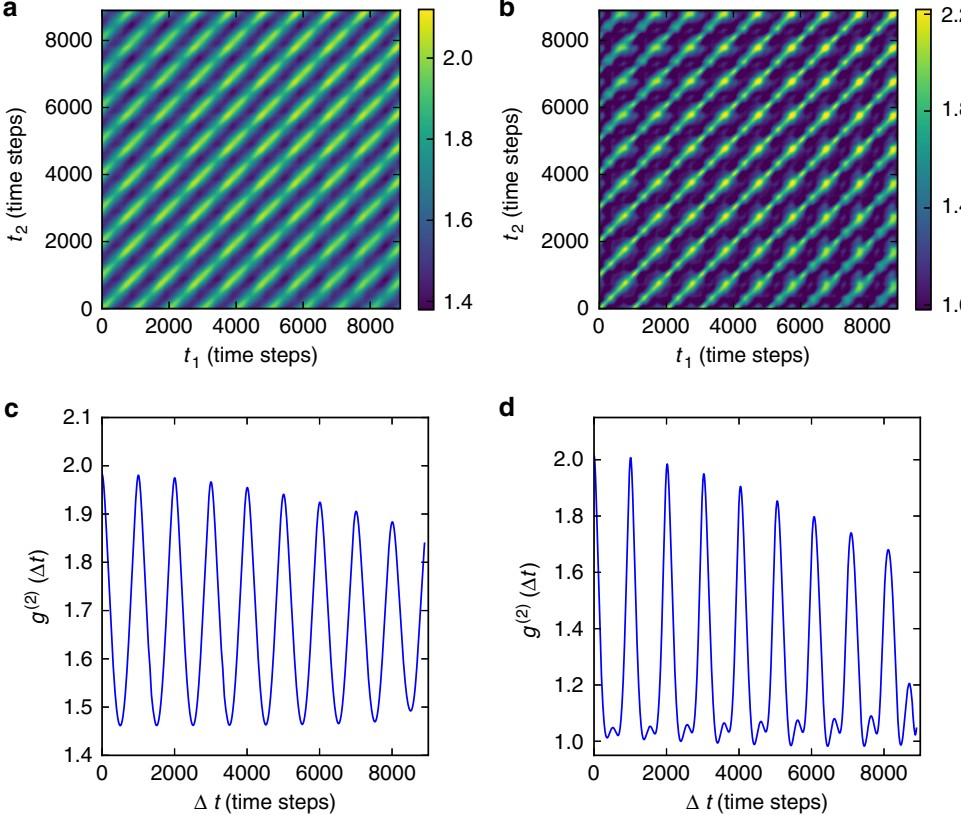

**Fig. 6** Numerical results for the Zeno model. **a** Two-time correlations calculated at 0.39 inverse lattice units, which corresponds to a length of 16 units. **b** Two-time correlations calculated at 0.91 inverse lattice units, corresponding to 7 units. **c** $g^{(2)}(\Delta t)$ autocorrelations at 0.39 inverse lattice units (**d**) $g^{(2)}$ $(\Delta t)$ autocorrelations at 0.91 inverse lattice units

characterizes the strength of step-edge barriers. The main equations of the model are shown in the inset of Fig. 5a. In our implementation of the model, the lattice unit $a$ is set to unity and the flux $F$ is set to 1/1000 so that the average height of the surface $h(t) = aFt$ advances by one unit for every 1000 time-steps. The mound is generated starting from a fixed base layer with a radius of 200 lattice units, and the critical length for nucleation of a new top layer is set at 10 lattice units. The Schwoebel length $L_s$ is defined as follows: if the width L of a terrace is smaller than $L_s$, most of the atoms landing on this terrace go to its upper edge, while if $L > L_s$, about one-half of the atoms go to each edge because they are too far from the other one. The velocity of each step is calculated from the lengths of the lower ($n$th) and upper ($n + 1$th) terraces. The step spacing becomes very small as the steps slow down near the edge of the mound. However, the overall shape of the island approaches a nearly stationary state. Figure 5b shows the surface configuration for the last 9 lattice units of deposition. The step configuration at different times is almost identical for times separated by an integer number of deposited layers (1000 time-steps). However, for curves separated by only 0.5 layers, it is seen that the steps advance halfway across their lower terrace, and this behavior is independent of the local terrace length. This is precisely the behavior that we require in order to explain the experimental data, where the step velocity is a variable proportional to the upper and lower terrace widths, but the period $T_{LSF}$ to advance by one step spacing is essentially fixed.

Figure 6 shows autocorrelations calculated for the LSF-Zeno model. Figure 6a is calculated using Eq. 2 over a range of $Q_{\|}$ that corresponds to a length scale slightly larger than the mean terrace width (~10 lattice units between 0 and 150 radius). The results exhibit parallel diagonal streaks that are separated by 1000 time-

steps, corresponding to integer monolayer time differences. An important feature of the results is that there are no strong modulations along the direction of the main diagonal, which indicates that correlations persist throughout the monolayer growth cycle. This indicates that the moment when nucleation occurs is not special as far as the correlations are concerned; only the time differences matter. This is the characteristic of stationary dynamics, i.e., where local fluctuations occur but the average structure does not change at an appreciable rate. In this case, the fluctuations take the form of an expanding array of concentric molecular steps.

As a result of the stationary dynamics, we can average over the age $[t_{age} = (t_1 + t_2)/2]$, to produce one-time correlations as a function of $\Delta t = (t_1 - t_2)$ for the LSF-Zeno model. An example is shown in Fig. 6c, which exhibits oscillations with a period of 1 layer (1000 time-steps). The frequency does not change for autocorrelations at different $Q_{\|}$, however weak harmonic beats are observed for $Q_{\|}$ values that corresponds to a length scale of approximately half of the mean terrace width in the central part of the mound (Fig. 6d). This behavior closely resembles the experimental data in Fig. 3b. In the Zeno model, the nucleation always occurs in the exact center of the top terrace. We note that random nucleation has been previously investigated in $1 + 1$ dimensions, and is found to produce more disordered step arrays[31], which would have the effect of increasing the relaxation rates in Eq. 4.

Several key features of the experimental results are reproduced by the LSF-Zeno model of molecular beam epitaxy described above. First, the oscillations in the correlation results are independent of the phase of the growth cycle. Second, the unexpected observation of a single oscillation period $T_{dep}$,

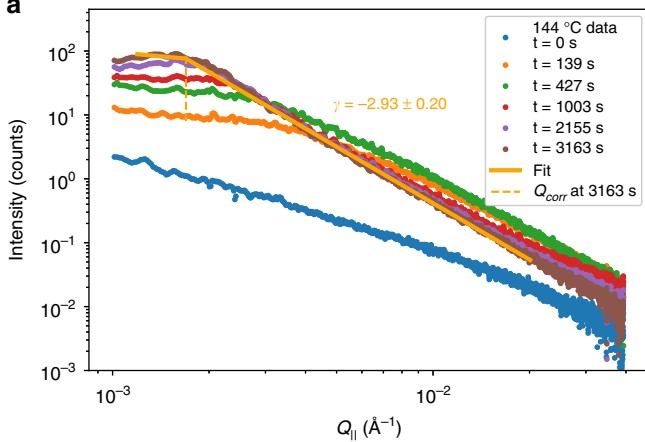

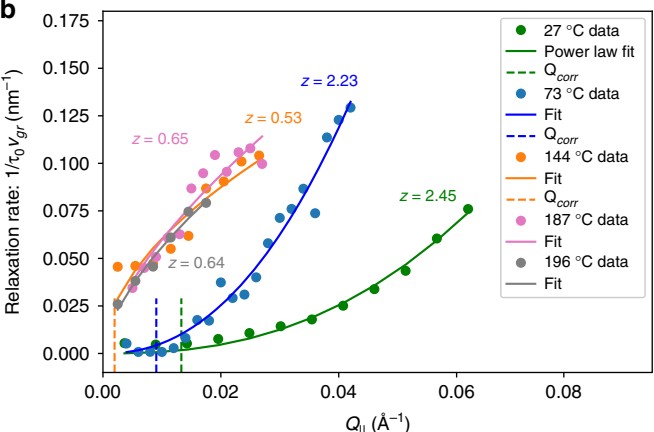

**Fig. 7** Scaling analysis of correlations and intensities. **a** Analysis of intensity data for a $C_{60}$ thin film deposited at 144 °C to extract the exponent $\gamma$. $Q_{corr}$ is the value of $Q_{\parallel}$ where the intensity breaks away from a power law dependence. **b** Correlation relaxation rates at different growth temperatures derived from $g^{(2)}(Q_{\parallel}, \Delta t)$ curves during steady-state growth. The curves are scaled by the growth velocity $v_{gr}$ in order to take into account differences in the deposition rates for different films. The dynamic exponent $z$ changes abruptly between the low-temperature and high-temperature regimes

independent of $Q_{\parallel}$ is found to be fundamental to local step flow with the specific step velocity distribution produced by the Zeno model. Third, the presence of mounds itself implies that nucleation occurs predominantly on the top terrace due to the fact that the mounds are a consequence of step-edge barriers leading to three-dimensional growth. The layer-by-layer process continues at the top of the mound, while the steps bounding lower terraces propagate by step flow. It is striking that all of these features of the LSF-Zeno model closely reproduce the experimental results.

**Scaling behavior.** Scaling relations can be useful for capturing the time- and wavelength-dependence of surface evolution during thin film deposition. Power-law behavior is frequently encountered, both for stable surfaces that exhibit kinetic roughening, as well as for unstable surfaces that exhibit mound or pattern formation[36,37]. We observe that $C_{60}$ thin film growth is of the second type, i.e., it is unstable to the formation of mounds. The unstable case includes examples of both growth by molecular beam epitaxy of elemental metals such as Pt on single-crystal Pt(111) substrates[29], as well as during sputter deposition of amorphous thin films of $Zr_{65}Al_{7.5}Cu_{27.5}$ alloys[38]. $C_{60}$ growth on randomly

oriented graphene domains falls into a third category of oriented polycrystalline thin films, which is a common mode for organic semiconductor materials when they are deposited on non-crystalline substrates.

In the early part of the growth of $C_{60}$ on a graphene-coated substrate, we observe that the mound structure evolves quickly. After an induction period, it gradually approaches a nearly static configuration that we refer to as the steady-state. In order to confirm the convergence to a static structure, we use the same X-ray scattering data used for the XPCS analysis by averaging over speckles to obtain the average intensity $\bar{I}(Q_{\parallel}, t)$. Figure 7a shows the intensity plotted as a function of $Q_{\parallel}$. In the early time, the scattering profile takes the form of a broad peak that is consistent with a long-wavelength surface instability[39]. It subsequently converges to a power-law form $\bar{I}(Q_{\parallel}) \propto Q_{\parallel}^{-\gamma}$ for $t \gtrsim 427\,s$, which corresponds to the time interval when the step-flow correlations appear in Fig. 2.

While the intensity profiles characterize the time-averaged mound structure, the correlations characterize fluctuations about the time-averaged structure. Step flow itself can be considered to be a fluctuation that produces periodic oscillations in the correlations. In addition, there can be significant fluctuations in the spacing and jaggedness of step edges because ad-molecules diffusing on the terraces attach at random positions[40], and because nucleation on the topmost terrace of the mounds is a stochastic process that does not occur precisely at the center[31,35]. These effects cause the correlations to decay monotonically as a function of time difference. In order to characterize the dynamics from coherent X–ray scattering data, we assume a scaling form for the steady-state dynamics:

$$\langle h(Q_{\parallel}, t_1) h(Q_{\parallel}, t_2) \rangle \sim g_{SS}(Q_{\parallel}^z |t_1 - t_2|) \quad (5)$$

where $h(q_{\parallel}, t)$ is the Fourier component of the surface amplitude at wave vector $Q_{\parallel}$ and time $t$[41]. This expression is valid for $t_1, t_2 \to \infty$ and $\Delta t = |t_1 - t_2|$ finite, i.e., the steady-state regime. Also, $\lim_{x \to \infty} g_{SS}(x) = 0$, so Eq. 5 is consistent with an intermediate scattering function that decays as $g^{(1)}(Q_{\parallel}, \Delta t) = I_0 \exp\{-(\Delta t / \tau_0)^n\}$ with $\tau_0(Q_{\parallel}) \sim Q_{\parallel}^{-z}$. We focus on the $Q_{\parallel}$ dependence of $1/\tau_0$ rather than the other time constants in Eq. 4 since it can be measured in both the high- and low-temperature growth regimes.

We have established in the previous section that the growth in the high temperature range is consistent with a step-flow model. This raises several interesting questions: do the dynamics obey the scaling relations in Eq. 5? Can a transition in the dynamics be observed from low deposition temperatures where local relaxation dominates, to higher temperatures where longer-range diffusion of molecules on terraces play a role? In order to investigate these questions, we have deposited $C_{60}$ thin films with the substrate held at temperatures below 100 °C, which results in very small grain size polycrystalline thin films. Figure 7b shows the relaxation rates extracted by XPCS analysis for substrate temperatures of 27° and 73 °C. The dynamic exponent extracted from fitting to this data is in the range $z = 2.2$–2.5. These exponents can be compared to those measured for amorphous sputter-deposited Si and $WSi_2$ thin films, $z = 1.2$ to 2.0[13,17]. The difference may be related to the fact that sputter-deposited Si and $WSi_2$ thin film surfaces exhibit characteristics of kinetic roughening, where roughening is driven by noise in the deposition flux. On the other hand, for $C_{60}$ the surface is unstable to mound formation due to deterministic processes, which may dramatically shift the exponents. For example, Krug predicts a dynamic scaling exponent $1/z = 1/4$ for unstable mound growth[37]. X-ray intensity profiles suggest that coarsening

of crystalline domains also plays a role in determining the exponent for $C_{60}$ films (Supplementary Note 6).

Figure 7b also shows a comparison with relaxation rates for $C_{60}$ films deposited at temperatures above 140 °C where larger mounds with well-defined step arrays are formed. In this range, the XPCS results during steady-state growth exhibit clear oscillations, as illustrated in Figs. 2 and 3. At higher temperatures, the relaxation is characterized by the overall time constant $\tau_0(Q_\parallel)$. The observed relaxation rates in Fig. 7b are significantly higher at temperatures >140 °C compared to low temperature deposition, and the dynamic exponent changes significantly, from $z = 2.2 - 2.5$ at lower temperatures to $z = 0.53 - 0.65$ at higher temperatures. These results clearly indicate a transition to a new regime.

This analysis reveals an important finding, that $z$ characterizes the non-equilibrium dynamics of step-flow in the steady-state regime. The unusually low value of $z \approx 0.6$ may be related to the fact that as $Q_\parallel$ increases, the correlations are most sensitive to steps with smaller terrace spacing, which propagate more slowly according to the step-flow model presented in the previous section. This is opposite to many typical situations, such as Brownian motion or kinetic roughening, where relaxation rates increase at shorter length scales. We also find that the oscillations in the correlations are not observable for growth at 73 °C and below, which indicates that correlated step-flow does not occur at lower temperatures. Thus, the large change in the dynamic exponent is linked to a fundamental change in the dynamics of the surface during growth.

## Discussion

In conclusion, coherent X-ray scattering is used to investigate step-edge motion on surfaces. A transition of the surface dynamics has been observed on growing $C_{60}$ surfaces that is related to a change from roughening dominated by ad-molecule diffusion at higher temperatures to a process at low temperatures where correlated step motion is absent. In the high-temperature regime, oscillations in correlations due to step-flow are observed as the surface returns to a self-similar state each time the steps advance by one terrace length. This information would be difficult to obtain by any other experimental technique, and it is completely invisible to methods based on low-coherence X-rays. The results illustrate a broadly applicable and powerful method with great promise for characterizing surface dynamics, and for a direct comparison with theoretical models of surface growth and fluctuations.

## Methods

**In-situ coherent X-ray experiments**. The experiments were performed at the National Synchrotron Light Source II, Coherent Hard X-ray beamline in a custom deposition chamber. $C_{60}$ is deposited from a thermal source onto thermally oxidized silicon substrates coated with single-layer graphene. The purpose of the graphene is to promote alignment of the thin films. The substrate temperatures are controlled between room temperature and 80 °C with a recirculating chiller/heater, while higher temperatures are achieved using a resistive heater embedded in the sample mount. The X-ray energy was 9.65 keV, with a coherent flux at the sample of $\sim 10^{11}$ ph/sec, and a focused beam size of $10 \times 10$ μm². The angle of incidence of X-rays on the sample was 0.4°. This low angle of incidence leads to an illuminated area of the surface that is elongated by a factor of 140 along the beam direction, so that the footprint of the X-ray beam on the surface is $1400 \times 10$ μm². X-ray diffuse scattering was monitored at a grazing exit angle of ~0.1°, significantly below the critical angle for total reflection in order to achieve good surface sensitivity. X-rays were detected with an Eiger 4M area detector at a rate of 2.5 fps. The detector has a pixel size of 75 μm, and it was placed at a distance of 10.2 m from the sample for these measurements. X-ray Photon Correlation Spectroscopy data analysis was performed by standard methods (Supplementary Note 1).

**Ex-situ characterization of thin film surfaces**. Post-deposition atomic force microscopy (Asylum MFP-3D) and X-ray diffraction (Bruker D8 Discover) was used to confirm that the films are polycrystalline with (111) orientation in all cases.

**Simulated mound scattering intensities and correlations**. X-ray structure factors and scattering intensities were calculated for a simulated mound growing according to the Zeno growth model. The base layer radius is fixed at $R = 200$, and all higher layers are constrained from exceeding that size. The total height $M$ of the mound increases as new layers nucleate at the mound's apex, and the radii $R_j(t)$ increase with time, representing the lateral propagation of steps. In this study, the growth was first propagated to $M = 400$ in order to establish a nearly stationary mound shape, and then an additional 10 layers were propagated with 1000 time-steps per layer in order to generate scattering intensities and correlations. See Supplementary Note 7 for details of the scattering intensity calculation for the simulated mound.

## Data availability

Data supporting the findings of this study are available within the article and its Supplementary Information file and from the corresponding author on reasonable request.

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

## Acknowledgements

We are grateful for the contributions of Darren Sullivan for assistance in developing the sample heating and thermal evaporator control software; to Jeff Bacon of the Boston University MSE Core Research Facility for assistance with the ex-situ X-ray diffraction analysis; to Rick Greene for technical support at the CHX beamline; and to Nicole Bouffard of the Microscopy Imaging Center at the University of Vermont for technical assistance with the AFM imaging. This material is based upon work supported by the U. S. Department of Energy (DOE) Office of Science under Grant No. DE-SC0017802. K.F. L. and P.M. were supported by the National Science Foundation (NSF) under Grant No. DMR-1709380. The Bruker X-ray diffractometer was acquired under NSF grant No. 1337471. This research used the 11-ID beamline of the National Synchrotron Light Source II, a U.S. DOE Office of Science User Facility operated for the DOE Office of Science by Brookhaven National Laboratory under Contract No. DE-SC0012704.

## Author contributions

R.L.H., K.F.L., and J.G.U. conceived and designed the experiments. R.L.H., Y.L., K.F.L., P. M., J.G.U., and J.W. participated in the coherent X-ray data collection. A.F., L.W., and Y. Z. developed the GISAXS instrumentation and the XPCS data reduction and correlation software. R.L.H. and J.G.U. constructed the growth chamber and analyzed the coherent X-ray data. K.F.L. and P.M. performed post-growth X-ray diffraction analysis. Y.L. performed post-growth AFM analysis. R.L.H. wrote the first draft of the manuscript. All authors discussed the results and reviewed the manuscript. R.L.H. supervised the project.

## Additional information

**Competing interests:** The authors declare no competing interests.

