## [Peer Review File · Nature Communications]

Reviewers' comments:

Reviewer #1 (Remarks to the Author):

Summary:

The authors of this work appear to have conducted an ambitious set of coherent x-ray scattering measurements during the growth of C60 layers upon a substrate at varying temperatures. The most novel portion of the work appears to be the x-ray correlation spectroscopy utilized to measure the growth modes of the C60 layers, which in the absence of direct imaging (which itself is not always/usually possible), can provide direct information about the growth modes of the molecules at the surface. In particular step-flow growth does not lend itself to direct observation, in real-time, using normal techniques. The novelty of the experiment is likely only possible through the use of increased brilliance of the NSLS-II light source, making this experiment impossible outside of a synchrotron environment, and likely very difficult at any other facility. These coherent x-ray measurements are then supplemented by conventional materials characterization techniques for supporting information, as well as numerical simulation/modeling.

The application of coherent x-rays to study growth modes is relatively new, though not entirely as the authors themselves cite/attribute. However, this appears to be a very high quality demonstration of the technique for atomic/molecular growth modes, likely the best to date. The technique of XPCS itself is well established.

Ordered atomic and molecular growth at an interface is an area of wide application with interest from a variety of fields, which I believe to be another point in favor of the manuscript.

Notes:

The temperature dependence demonstrated in the manuscript is very nice (both time constants/relaxation rates, and terrace lengths). However it seems like there might be plausible that the step-flow frequency itself might be temperature dependent in a meaningful fashion, instead of connecting it through terrace size. I can't seem to find that in the manuscript. If it is the case that this should be something universal, regardless of dimensions, then I would request that the authors articulate this better.

I believe some mention of the experimental detail should be moved to earlier in the primary manuscript from the late/extra information. The manuscript itself does not readily indicate that this is a set of small/glancing angle scattering measurements, unless one is familiar with inverse length units, nor is it readily clear (in the primary manuscript) that this is a synchrotron based technique and not one generally available/applicable from some sort of lab based SAXS machine. This novelty should come through on its own. Perhaps just even a scattering pattern included in the main paper would help.

Why does the speckle contrast appear to change so much? Is this Q-dependence, growth temperature dependent, or a combination? I do not understand this and I believe it deserves at least some mention. Is this an indication of dynamics that are proceeding faster than is being measured?

They note the beam size as $10\mu\text{m}^2$. I assume this is the beam cross section and not the size of the illuminated area. Likewise the AFM scans appear quite small. It might be useful to include approximate dimensions of the illuminated area to demonstrate how much of an ensemble average is being conducted.

Figure S2: please do not mix units within a single set of inset figures. Likewise can the time axes for the different insets be made to a uniform scale to enable easy comparison?

Would it be appropriate to show additional ex-situ microscopy from the growth at different (lower) temperatures to demonstrate the absence/difference of the smooth atomic terraces in the supplemental information?

Is it possible for an example of the speckle to be included in the supplemental information?

Figure 5: please give some color indication for the top-two panels so that a reader may understand the relation to correlation amount. This is needed for its own sake, but particularly as the color scheme from the experimental data in figure 1 is different.

Reviewer #2 (Remarks to the Author):

This MS reports on static and dynamic measurements, using coherent X-ray scattering, of the growth of C60 films on single layer graphene films on top of thermally oxidized silicon substrates during deposition at various temperatures. The authors claim agreement with a particular model of step flow during growth (the so-called Local Step Flow model) to explain their data, and also extract growth exponents at various temperatures.

The experimental data that is presented appears to be of high quality, and the model they have used to explain it is intriguing. Nevertheless, I have several issues with this MS which I list below.

(1) There are confusing aspects to the way the data is presented. For instance, on P.4, they state that there are “no oscillations in the diffuse scattering intensity” and refer to their fig. S6 in the Supplementary Information section, which looks at first glance like a smooth function of time, with some noise. But this region of q -space is exactly the region where their oscillatory time-dependent correlation functions, such as shown in Fig. S5 or Fig. 2 in the main text, are obtained from. It is hard to imagine a highly regular oscillatory periodic function of time emerging from the autocorrelation of a smooth function, so I am wondering whether the diffuse scattering does indeed show fine structure oscillations with time, and that the authors simply mean that it does not have the same periodicity as that of the reflectivity shown in Fig. S6.

(2) They refer to a $Q(\text{parallel})$ where the amplitude of the oscillations is largest as related to the inverse of the average step spacing. However, their time-averaged intensity vs $Q(\text{parallel})$ does not show a peak (even a broad one) which would confirm this. Unless they are referring to the Q value they have indicated by Q_{corr} in Fig. 6, which they make no further comment on. Perhaps they can claim that the $S(Q)$ shown in Fig.6 represents the “mounds” and not the steps, but one would think that it would show some structure from the steps at the kind of periodicities shown in structures such as shown in Fig. 4.

(3) They seem to adopt a one-dimensional model where the steps all flow in a single direction. However, they are depositing on a macroscopic surface, so a coherent X-ray beam of size 10μ (and a much larger footprint on the surface at grazing incidence) would probably see steps moving in random directions. This alone would be enough to give rise to an oscillatory g_2 function from homodyne scattering, even though uniform step motion in one direction might not.. However, even in the latter case, the fixed boundaries of the beam illuminating the surface, can give rise to beautifully regular oscillations (very similar to those seen here) in homodyne scattering, as steps move across the surface with uniform velocity, as has been already demonstrated in Ref. 12 of this MS.

(4) In the section on “Scaling” in Fig. 6, their $S(q)$ presumably demonstrates that the scattering is from a self-affine structure on the surface (e.g. from the “mounds”) but no attempt is made to relate the exponent they find for the decrease in $S(q)$ and the self-affine roughness exponent H , or to test the scaling law relationships between the various parameters such as z , H , n and dimension d , although they appeal to the KPZ scaling theory for deposition growth.

(4) Finally, and perhaps most importantly, it is unclear what they are showing in Fig. 5, when they are trying to show consistency of their model with the purely empirical formula (Eq. 4) they have

used to fit the data. I understand their model, but I don't really understand how they have calculated the g_2 function corresponding to a multiplicity of steps all moving with slightly different velocities and different spacings. For this they would at least have had to calculate the scattering from this kind of surface as a function of Q and t , which would be a complicated analytical calculation, or perhaps even a numerical simulation. I was hoping that there would be some description of how the correlation functions were calculated in the Supplementary Information, but no details are given except for the cartoon describing the simple one-dimensional model. If their model is taken literally, their Figs. 5 (c) and (d) would imply that the whole structure coherently reproduces itself periodically as a function of time with a period of one monolayer growth time, although the model is statistical.

It is possible that the authors have clearer explanations to these points. However, in view of the above points, I cannot recommend publishing this MS in Nature Communications.

Reviewer #3 (Remarks to the Author):

This manuscript reports the use of photon correlation spectroscopy from coherent x-ray scattering to investigate the motion of surface steps during epitaxial crystal growth. Understanding the source of the observed time correlations is non-trivial and the authors present a detailed analysis that connects their measurements with the motion of individual surface steps. The investigation represents a new application of XPCS and the analysis established in the manuscript provides a foundation for expanding this work. It is worth noting that the evolution of crystalline surface steps is central to understanding epitaxial growth so that the present study, combined with the development of new coherent x-ray sources, could lead to new and useful research tools in the future. I strongly support the publication of this manuscript in Nature Communications.

I have a few comments that I would like the authors to address.

1) In the discussion about determining the origin of the time correlations, it was stated that there is no obvious quasi-static reference signal. I would argue that there are quasi-static sources from large-scale features such as the mounds. So, for example, I could envision whole groups of steps flowing (as is the typical picture for step-flow growth) and the coherent beam would detect changes in phase (relative to the mound boundaries) from the moving group of steps. An argument to rule out this scenario is to consider the length-scale from the Q that gives the maximum amplitude of oscillations, which they give as approx. 110 nm. Looking at the AFM images, that is the approximate length scale of the surface steps. In other words, getting a length scale from the Q_{\max} (the Q for which I_1 of eq.4 is maximum) is a way to identify the feature being probed – in this case it's an individual step length.

The authors should give the step length from the AFM. It is very hard to see unless one expands the image on the computer screen. Some discussion of these points in the manuscript would be useful.

2) The discussion of the z exponent from Fig. 6 is a bit misleading. First, the KPZ equation and the idea of universality classes are outdated. These models assume deposition flux noise-driven kinetic roughening whereas real systems are somewhat deterministic with the noise coming from statistical fluctuations of the surface-evolution. There is a wealth of more recent kinetic Monte Carlo studies which show mounding and these give a broad range of exponents. The authors are correct to point out that there is a transition in the z value going into the step flow regime and it is a nice result to show. But we would not expect the z value in the step flow regime to be related to kinetic roughening models of the type they have referenced.

3) Fig. S1(a) can be a little confusing at first. While it is mathematically correct to call the vertical

axis Q_z , there is an alternative and equally correct designation as a surface-parallel direction, say Q_x , which better describes what we are seeing the figure. Specular reflection only occurs at one point and the rest of the vertical streak is actually probing the diffuse scattering along the surface (but orthogonally to their Q_{parallel} plotted on the horizontal axis). Due to geometrical factors, if one were to give in-plane Q_x values to the vertical streak, these values would be tiny compared to Q_{parallel} on the horizontal axis (as well as compared to values of Q_z). It would be, therefore, useful if the authors could present a schematic diagram of the experimental setup and show the orientation of Fig. S1(a) relative to it. I think it would help to re-label the vertical axis of Fig.S1(a) to be the other in-plane direction, although I will leave it to the authors' discretion as there could be other unintended consequences. At the very least, it will be useful to discuss the in-plane nature of the vertical streak and the associated in-plane Q -scale, which is very different than the scale for the Q_{parallel} plotted on the horizontal axis.

4) Fig. 6 should be given in inverse Angstroms rather than inverse nm because everything else in the manuscript is given in inverse Angstroms.

--- Responses to reviewers and changes in manuscript NCOMMS-18-30423-T ---

We would like to thank the referees for their careful reviews of the manuscript. We have taken into account all remarks and modified the manuscript accordingly. Please find below the detailed answers to specific queries, as well as the changes in the text.

We have also uploaded an annotated version of the manuscript highlighting changes in the text as a separate file. Additions and changes are in blue with a wavy underline, and deleted text is in ~~red with strikethrough~~. See the file 'C60_V8m_annotated.pdf'.

--- Response to Reviewer 1 ---

We have extracted specific comments from the "Notes" section of Reviewer 1's comments, which we address below. The excerpts of the reviewer's comments are shown in ***bold italics***.

...it seems like there might be plausible that the step-flow frequency itself might be temperature dependent in a meaningful fashion, instead of connecting it through terrace size.

Response: Steps can only flow through the addition of molecules arriving at the surface, and we don't expect any temperature dependence of the frequency. The only exception is that at high enough surface temperatures there can be desorption from the surface, which would cause the step velocity to be reduced, or even to reverse direction. We have not observed this latter effect in the experiments reported here, so we have not discussed it in the manuscript.

I believe some mention of the experimental detail should be moved to earlier in the primary manuscript from the late/extra information.

Response: We have inserted a new Fig. 1 and re-numbered the remaining figures. A new paragraph has also been inserted, starting at line 39 in the revised manuscript.

Why does the speckle contrast appear to change so much? ... Is this an indication of dynamics that are proceeding faster than is being measured?

Response: We have considered this question at length since it would be quite interesting if there were dynamics that proceed on faster timescales. However, we have concluded that there are other effects that account for the variation. The dependence on $Q_{||}$ occurs mainly due to the finite longitudinal coherence of the X-rays ($\Lambda \approx 0.13 \mu\text{m}$ at 0.1% BW) and the very long footprint of the scattering volume along the direction of the X-ray beam ($W \approx 1.4 \text{ mm}$ at $\alpha_i = 0.4^\circ$). In this case the optical path difference δ is dominated by the term $2W\sin^2(\psi)$, where ψ is the in-plane scattering angle. The contrast should be reduced by a factor $\exp(-2|\delta|/\Lambda)$, which evaluates to a 23% reduction of the optical contrast for $Q_{||} = 0.0235 \text{ \AA}^{-1}$. This $Q_{||}$ dependence is in reasonable agreement with the data shown in Fig. 3(a) of the revised manuscript. A second effect that we are aware of is that mechanical vibrations of the vacuum

pumping system reduce the contrast by varying amounts for different data sets. Fig. 3(b) of the revised manuscript has a reduced contrast due to this effect. We can diagnose this effect in the raw images since the speckles are slightly blurred, indicating that the sample changes its angle very slightly at a frequency higher than the data collection rate. This effect does not change the conclusions discussed in the manuscript. We are also working to reduce vibrations in future versions of the experiment in order to eliminate this artifact.

The data shown in the supplementary information exhibits an additional effect. Fig. S2(c) exhibits a larger $Q_{||}$ dependence than we expect based on the path length difference argument mentioned above. This is from shot noise due to the low signal at higher $Q_{||}$. It is more prevalent in the higher deposition temperature data, where the diffuse scattering is more compact along $Q_{||}$ due to the larger mound sizes.

It might be useful to include approximate dimensions of the illuminated area to demonstrate how much of an ensemble average is being conducted.

Response: This information has been added in the Methods section.

Figure S2: please do not mix units within a single set of inset figures. Likewise can the time axes for the different insets be made to a uniform scale to enable easy comparison?

Response: Figs. S2 (c), (e), and (f) have been updated to make the units and the range of time delays plotted consistent.

Would it be appropriate to show additional ex-situ microscopy from the growth at different (lower) temperatures to demonstrate the absence/difference of the smooth atomic terraces in the supplemental information?

Response: Additional AFM images for a thin film deposited at a lower temperature have been added to Fig. S4.

Is it possible for an example of the speckle to be included in the supplemental information?

Response: An example of the speckle pattern is shown in the new Fig 1, panel (a).

Figure 5: please give some color indication for the top-two panels so that a reader may understand the relation to correlation amount.

Response: Fig. 5 is now Fig. 6 in the revised manuscript. The color bars have been added in Figs. 6(a) and (b).

--- Response to Reviewer 2 ---

(1) It is hard to imagine a highly regular oscillatory periodic function of time emerging from the autocorrelation of a smooth function, so I am wondering whether the diffuse scattering does indeed show fine structure oscillations with time, and that the authors simply mean that it does not have the same periodicity as that of the reflectivity shown in Fig. S6.

Response: We have added revisions that emphasize that we are autocorrelating speckle intensities that are not a smooth function of time. Specifically, we have added a paragraph to the manuscript on pg. 5 that explains that individual speckles can oscillate, but these intensity oscillations disappear when averaging over a region of Q containing many speckles. In principle, we can plot the intensity of a single speckle to recover the oscillations, but the statistics are too limited in the present experiments to clearly observe that effect. We have also added a zoomed plot as part (b) of Supplementary Fig. S4 that demonstrates that there are no intensity oscillations in a region of the scan where oscillations in the correlations are very strong.

(2) Perhaps they can claim that the $S(Q)$ shown in Fig.6 represents the “mounds” and not the steps, but one would think that it would show some structure from the steps at the kind of periodicities shown in structures such as shown in Fig. 4.

Response: The steps are too disordered to exhibit peaks that correspond to the step spacing. We address this question in the new supplementary section S7, where weak peaks can be observed for a simulated mound. The simulated mound is significantly more ordered than what we observe in the experiment since the model does not include asymmetry of the mound or jaggedness of the step edges, which tends to reduce these peaks.

(3) They seem to adopt a one-dimensional model where the steps all flow in a single direction. However, they are depositing on a macroscopic surface, so a coherent X-ray beam of size 10μ (and a much larger footprint on the surface at grazing incidence) would probably see steps moving in random directions. This alone would be enough to give rise to an oscillatory g^2 function from homodyne scattering, even though uniform step motion in one direction might not. However, even in the latter case, the fixed boundaries of the beam illuminating the surface, can give rise to beautifully regular oscillations (very similar to those seen here) in homodyne scattering, as steps move across the surface with uniform velocity, as has been already demonstrated in Ref. 12 of this MS.

Response: We note that even though the Zeno model is one-dimensional, the steps moving in one direction have counterparts on the opposite side of the mound that move in the opposite direction. In the calculation of the structure factor in the model, we assume 2D symmetric circular steps all flowing out from the center of the mound, each with a time-dependent radius given by the one-dimensional Zeno model equations. These details of the model are addressed in the new supplementary section S7. While steps moving in opposite directions should lead to

oscillations, as the reviewer states, the frequency would be doubled compared to what we observe at Q_{\max} . If we add an extra term proportional to $\exp(-i\omega_1 t)$ to Eq. 4 to represent steps flowing in the negative direction, it produces an extra small modulation of the g_2 function proportional to $\cos^2(\omega_1 t)$ that is probably too weak to easily observe. It is possible that this effect contributes to the $2\omega_1$ term in Eq. 4, but it is not clear why it should be observable only at higher Q_{\parallel} . We also note that Fig. 1(c) in the revised manuscript illustrates how heterodyne mixing between scattering from the overall mound structure and the steps produces a large oscillatory signal, even though the scattering from the steps separately would be very weak.

Ref. 12 reports on coherent X-ray scattering from a Pt(001) surface with very large terraces. This is a very different situation than the current results since in Ref. 12 there is only at most a single step edge in the illuminated area at any given time. In this case of Ref. 12, the “speckle” pattern corresponds to either one speckle (for no step) or two speckles (for one step). However, in our case the surface is much more complex. The number of steps in the illuminated area is very large. Most of the steps never reach the edge of the illuminated area, and they are moving in different directions. Therefore, it is hard to see how the strong oscillations that we observe could possibly be due to steps entering and leaving the illuminated area.

(4) In the section on “Scaling” in Fig. 6, their $S(q)$ presumably demonstrates that the scattering is from a self-affine structure on the surface (e.g. from the “mounds”) but no attempt is made to relate the exponent they find for the decrease in $S(q)$ and the self-affine roughness exponent H , or to test the scaling law relationships between the various parameters such as z , H , n and dimension d , although they appeal to the KPZ scaling theory for deposition growth.

Response: Since mound formation has a preferred length scale it cannot be self-affine, and the scaling is not of the universal type. Please see our response to Reviewer 3 below for further comments on this issue.

(5) I was hoping that there would be some description of how the correlation functions were calculated in the Supplementary Information, but no details are given except for the cartoon describing the simple one-dimensional model. If their model is taken literally, their Figs. 5 (c) and (d) would imply that the whole structure coherently reproduces itself periodically as a function of time with a period of one monolayer growth time, although the model is statistical.

Response: See the new supplementary section S7, where we give some details of the model calculations. The model is purely deterministic, and it does lead to the whole structure being nearly reproduced for each monolayer deposited. This is shown explicitly in Fig. 5 of the revised manuscript (which is unchanged from the original manuscript aside from the figure number being increased by one).

--- Response to Reviewer 3 ---

1a) In the discussion about determining the origin of the time correlations, it was stated that there is no obvious quasi-static reference signal. I would argue that there are quasi-static sources from large-scale features such as the mounds. So, for example, I could envision whole groups of steps flowing (as is the typical picture for step-flow growth) and the coherent beam would detect changes in phase (relative to the mound boundaries) from the moving group of steps.

Response: This is a useful insight and we have adopted the idea in the revised manuscript. It is also consistent with the new Fig. 1, which emphasizes heterodyne mixing between the scattering from the average mounds and the moving steps.

1b) An argument to rule out this scenario is to consider the length-scale from the Q that gives the maximum amplitude of oscillations, which they give as approx. 110 nm. Looking at the AFM images, that is the approximate length scale of the surface steps. In other words, getting a length scale from the Q_{\max} (the Q for which I_1 of eq.4 is maximum) is a way to identify the feature being probed – in this case it's an individual step length.

Response: We have added the requested information to the manuscript. The length scale of the feature being probed (126 nm from Q_{\max}) is close to the estimated mean step spacing (150 nm from AFM). Thus, as Reviewer 3 points out, the feature being probed is the step spacing.

2) The discussion of the z exponent from Fig. 6 is a bit misleading. First, the KPZ equation and the idea of universality classes are outdated. These models assume deposition flux noise-driven kinetic roughening whereas real systems are somewhat deterministic with the noise coming from statistical fluctuations of the surface-evolution. There is a wealth of more recent kinetic Monte Carlo studies which show mounding and these give a broad range of exponents. The authors are correct to point out that there is a transition in the z value going into the step flow regime and it is a nice result to show. But we would not expect the z value in the step flow regime to be related to kinetic roughening models of the type they have referenced.

Response: The comparison with KPZ growth was intended to be with the z exponents from an earlier paper on deposition of amorphous Si and WSi_2 thin films, which do exhibit characteristics of kinetic roughening and self-affine growth. Since KPZ is not directly relevant to the present results, we have deleted the reference to it in order to avoid confusion.

3) Fig. S1(a) can be a little confusing at first. While it is mathematically correct to call the vertical axis Q_z , there is an alternative and equally correct designation as a surface-parallel direction, say Q_x , which better describes what we are seeing the figure. Specular reflection

only occurs at one point and the rest of the vertical streak is actually probing the diffuse scattering along the surface (but orthogonally to their Q_{parallel} plotted on the horizontal axis). Due to geometrical factors, if one were to give in-plane Q_x values to the vertical streak, these values would be tiny compared to Q_{parallel} on the horizontal axis (as well as compared to values of Q_z). It would be, therefore, useful if the authors could present a schematic diagram of the experimental setup and show the orientation of Fig. S1(a) relative to it. I think it would help to re-label the vertical axis of Fig.S1(a) to be the other in-plane direction, although I will leave it to the authors' discretion as there could be other unintended consequences. At the very least, it will be useful to discuss the in-plane nature of the vertical streak and the associated in-plane Q -scale, which is very different than the scale for the Q_{parallel} plotted on the horizontal axis.

Response: A comment has been added in the supplementary information that the maximum value of Q_x in the image is $Q_x = 1.1 \times 10^{-4} \text{ \AA}^{-1}$, which occurs at the lower end of the central streak, so that the minimum length scale probed along the central streak is $\approx 55,000 \text{ \AA}$ ($5.5 \mu\text{m}$). Since that is larger than the mound size, there is little useful information about the in-plane surface structure along the vertical direction. We also note that Q_x varies in both the horizontal and the vertical direction, so that it is not possible to label the vertical axis of the image with a unique value of Q_x unless we distort the image for this purpose.

4) Fig. 6 should be given in inverse Angstroms rather than inverse nm because everything else in the manuscript is given in inverse Angstroms.

Response: The units have been changed to inverse angstroms. Fig. 6 is now Fig. 7 in the revised manuscript.

--- Changes in the revised manuscript ---

The numbers in square brackets below correspond to the line numbers of highlighted changes in the annotated revised manuscript, which is uploaded as a separate file. Please note that the figures in the annotated version of the manuscript are at a reduced resolution in order to reduce the file size.

- 1) [39-55 and new Fig. 1] This addition includes an example of the scattering pattern, some experimental details, an example of the speckle pattern, a diagram of the experimental setup, and a diagram that shows the orientation of Q_{\parallel} relative to the surface. We have also emphasized the heterodyne principle, which is useful for understanding the origin of the oscillatory correlations.
- 2) [Fig. 3 caption] It is noted that one of the curves in Fig. 3 has been shifted upwards by 0.01 to keep it from overlapping with the other curve. This information was previously missing, which gave the impression that the speckle contrast changes faster with Q_{\parallel} than it actually does.

- 3) [91-92] The average step spacing has been added, as well as a comparison to the length scale calculated from Q_{\max} .
- 4) [107] The word “heterodyne” has been added. We are emphasizing heterodyne mixing in the new version of the manuscript.
- 5) [120-121] We have adopted the point of view proposed by Reviewer 3, that the scattering from slowly-evolving larger features such as the mounds can be considered to be a quasi-static reference signal. A sentence has been added to that effect.
- 6) [128-134] A paragraph has been added that describes in an intuitive way how oscillations in the correlations can be observed when there are no intensity oscillations.
- 7) [146-160] The discussion has been changed to a description in terms of heterodyne mixing.
- 8) [173-174] It is pointed out that the model is deterministic.
- 9) [180-181] An additional detail of the model has been added.
- 10) [259-271] All mentions of KPZ and universality have been removed.
- 11) [296-299] The beam size is stated clearly, and the dimensions of the illuminated area are given.
- 12) [307-309] A data availability statement has been added.
- 13) [431-432] Ref. 42 (Kardar, Parisi, and Zhang) has been removed.

--- Changes in the Supplementary Information ---

- 1) Section S1: New information has been added in the second paragraph describing the grain size extracted from the $Q_{||}$ position of the vertical streaks in Fig. S1(a), and the length scales probed by the variation of Q_x along the central streak where $Q_y = 0$.
- 2) Section S2: Figs. S2 (c), (e), and (f) have been updated to make the units and the range of time delays plotted consistent. New information has been added to the first paragraph of the section describing artifacts in the data that tend to reduce the speckle contrast.
- 3) Section S3: Two new AFM images have been added to Fig. S4. They are for a lower temperature deposition (106°C) where surface steps are not readily visible.
- 4) Section S4: There were no changes in this section.
- 5) Section S5: The region between $t=1600$ and 2400 s in Fig. S6(a) has been marked by a dashed line, and a new panel [Fig. S6(b)] has been added showing a close-up view of this region of the intensity curve.
- 6) Section S6: There were no changes in this section.
- 7) Section S7: This section, including Fig. S8 are new in this revised version of the supplementary information file.

REVIEWERS' COMMENTS:

Reviewer #1 (Remarks to the Author):

I believe the authors have addressed the major concerns of the referees in adequate fashion. They have presented their rebuttal in a clear and transparent fashion, and most importantly addressed these questions in the text of the manuscript.

Their statements, made through the manuscript itself, do now provide a consistent model for the observed behavior. While it might be possible in the future to provide more than consistency, I believe at this juncture that the authors have sufficiently and successfully defended their conclusions, going from the realm of possible and plausible, to one of likely true. Or said differently, any different interpretation of the results would need to overcome the significant hurdle of being able to account for the observed behavior of the steps better than the model/interpretation presented by the authors.

All minor corrections to the manuscript have been sufficiently made.

Reviewer #2 (Remarks to the Author):

I believe the authors have answered my original doubts concerning this MS, and clarified several things in the revised MS. I thus favor publication of this MS in Nature Communications.

Reviewer #3 (Remarks to the Author):

The authors have diligently responded to my comments as well as those of the other reviewers. I support the publication of the manuscript in Nature Communications.

Response to reviewers:

The reviewers recommended the manuscript for publication without requesting any further changes. There were no issues raised. Therefore, no response is needed.

REVIEWERS' COMMENTS:

Reviewer #1 (Remarks to the Author):

I believe the authors have addressed the major concerns of the referees in adequate fashion. They have presented their rebuttal in a clear and transparent fashion, and most importantly addressed these questions in the text of the manuscript.

Their statements, made through the manuscript itself, do now provide a consistent model for the observed behavior. While it might be possible in the future to provide more than consistency, I believe at this juncture that the authors have sufficiently and successfully defended their conclusions, going from the realm of possible and plausible, to one of likely true. Or said differently, any different interpretation of the results would need to overcome the significant hurdle of being able to account for the observed behavior of the steps better than the model/interpretation presented by the authors.

All minor corrections to the manuscript have been sufficiently made.

Reviewer #2 (Remarks to the Author):

I believe the authors have answered my original doubts concerning this MS, and clarified several things in the revised MS. I thus favor publication of this MS in Nature Communications.

Reviewer #3 (Remarks to the Author):

The authors have diligently responded to my comments as well as those of the other reviewers. I support the publication of the manuscript in Nature Communications.